# Effect of Non-Surgical Periodontal Treatment on Oxidative Stress Markers in Leukocytes and Their Interaction with the Endothelium in Obese Subjects with Periodontitis: A Pilot Study

**DOI:** 10.3390/jcm9072117

**Published:** 2020-07-04

**Authors:** Mayte Martínez-Herrera, Zaida Abad-Jiménez, Francisco Javier Silvestre, Sandra López-Domènech, Javier Silvestre-Rangil, Cecilia Fabiana Márquez-Arrico, Víctor M. Víctor, Milagros Rocha

**Affiliations:** 1Department of Stomatology, University of Valencia, Gascó i Oliag 1, 46010 Valencia, Spain; maytemartinez05@gmail.com (M.M.-H.); francisco.silvestre@uv.es (F.J.S.); cecilia.fabiana.m@gmail.com (C.F.M.-A.); 2Department of Stomatology, University Hospital Doctor Peset-FISABIO, Av. Gaspar Aguilar 90, 46017 Valencia, Spain; javier.silvestre@uv.es; 3Department of Endocrinology and Nutrition, University Hospital Doctor Peset-FISABIO, Av. Gaspar Aguilar 90, 46017 Valencia, Spain; zaiaji@alumni.uv.es (Z.A.-J.); Sandra.Lopez@uv.es (S.L.-D.); 4CIBER CB06/04/0071 Research Group, CIBER Hepatic and Digestive Diseases, University of Valencia, Av. Blasco Ibáñez 15, 46010 Valencia, Spain; 5Department of Physiology, University of Valencia, Av. Blasco Ibáñez 15, 46010 Valencia, Spain

**Keywords:** periodontitis, reactive oxygen species, oxidative stress, endothelial dysfunction, periodontal treatment, dietary therapy, obesity

## Abstract

Aim: The primary objective of this pilot study was to evaluate the effect of non-surgical periodontal treatment. The secondary aim was to evaluate the effect of dietary therapy on both parameters of oxidative stress in leukocytes and leukocyte-endothelial cell interactions in an obese population. Methods: This was a pilot study with a before-and-after design. Forty-nine obese subjects with periodontitis were randomized by means of the minimization method and assigned to one of two groups, one of which underwent dietary therapy while the other did not. All the subjects underwent non-surgical periodontal treatment. We determined periodontal, inflammatory and oxidative stress parameters—total reactive oxygen species (ROS), superoxide production, intracellular Ca^2+^, mitochondrial membrane potential and superoxide dismutase (SOD) activity. We also evaluated interactions between leukocytes and endothelium cells—velocity, rolling flux and adhesion—at baseline and 12 weeks after intervention. Results: Periodontal treatment improved the periodontal health of all the patients, with a reduction in serum retinol-binding protein 4 (RBP4), total superoxide production and cytosolic Ca^2+^ in leukocytes. In the patients undergoing dietary therapy, there were less leukocyte adhesion to the endothelium, an effect that was accompanied by a decrease in TNFα, P-selectin and total ROS and an increase in SOD activity. Conclusions: Whereas non-surgical periodontal treatment induces an improvement in leukocyte homeostasis, dietary therapy as an adjuvant reduces systemic inflammation and increases antioxidant status which, in turn, modulates leukocyte-endothelium dynamics.

## 1. Introduction

Obesity is a chronic inflammatory disease associated with a series of comorbidities, including dyslipidemia, arterial hypertension, diabetes mellitus and periodontitis [1,2]. Obesity and periodontitis share an inflammatory component in their pathophysiology. Periodontitis is a chronic multifactorial inflammatory disease associated with dysbiotic plaque biofilms and characterized by host-mediated inflammation that results in the progressive destruction of periodontal attachment [3]. The host cells’ first defense strategy against periodontal pathogens is to release proinflammatory cytokines, which stimulates infiltration of polymorphonuclear leukocytes (PMNs). Prolonged inflammation as a response to bacterial plaque causes leukocytes to generate reactive oxygen species (ROS) such as hydrogen peroxide and superoxide. Together with disrupted antioxidant defenses, this leads to oxidative stress and apoptosis in the periodontal tissue [4,5], suggesting there are shared mechanisms in periodontitis and systemic inflammatory diseases. In fact, oxidative stress occurs in periodontal tissues, gingival crevicular fluid, saliva, serum and plasma [6]. In spite of this, very little research has centered on the production of ROS by PMNs under unstimulated conditions [7,8,9]. Mitochondria are the most important source of ROS in cells [10] and are essential for several cellular functions, including ATP production, intracellular Ca^2+^ regulation and cell survival. The overproduction of ROS can induce several maladaptive responses that stimulate mutagenesis and the inflammatory response. Diminished mitochondrial biogenesis, altered membrane potential, genetic factors and aging simultaneously contribute to mitochondrial dysfunction, which appears to be a central cause of insulin resistance and cardiovascular complications [11], though little is known about its role in periodontal disease.

There is a close interrelation among oxidative stress, mitochondrial dysfunction and inflammation, and all are key players in the pathogenesis of atherosclerosis [12]. Endothelial dysfunction, which leads to atherosclerosis, is initiated by the accumulation of leukocytes in the vessel wall and their eventual movement into the subendothelial space. This process is mediated by cell adhesion molecules on white blood and/or endothelial cells, whose expression is enhanced in these circumstances [13]. In this context, retinol-binding protein 4 (RBP4)—an adipokine primarily secreted by adipocytes and hepatocytes and involved in transporting retinol in blood—has been associated with endothelial dysfunction [14], cardiovascular disease [15] and periodontitis [16,17].

Epidemiological studies suggest that periodontitis plays a role in atherosclerotic cardiovascular diseases [18,19]. In fact, periodontitis and endothelial dysfunction have been shown to be closely connecting [20,21]. In line with this, we have shown in a previous study that the release of ROS from leukocytes is exacerbated in a proinflammatory state, thus contributing to oxidative stress and endothelial dysfunction in obesity and periodontitis [8,22]. Furthermore, we have demonstrated worse clinical outcomes in obese patients than in non-obese subjects after non-surgical periodontal treatment (at 3 months) [17], which is evidence of a negative effect of obesity on the response to periodontal treatment. In this sense, therapy for periodontitis in obese patients is likely to be more beneficial if there is weight loss, as it reduces levels of systemic proinflammatory cytokines [23,24]. However, the effect of periodontal treatment on ROS generation and on atherosclerotic markers in patients with periodontitis has received little attention by researchers [7,25,26]. Therefore, the primary objective of the current pilot study—which had a before-and-after design—was to explore the effect of non-surgical periodontal treatment on parameters of oxidative stress in leukocytes and leukocyte-endothelial cell interactions in an obese population with periodontitis. As a secondary aim, we set out to determine whether adjunctive dietary therapy can modulate these responses.

## 2. Materials and Methods

### 2.1. Subjects

This pilot study was conducted at University Hospital Dr. Peset (Valencia, Spain), from October 2018 to January 2020, and followed a before-and-after design. Obese patients were recruited from the Department of Endocrinology and Nutrition and the Department of Odontology and met the following criteria: Age between 30 and 60 years old; obesity (body mass index (BMI) ≥30 kg/m^2^, according to the criterion of the Spanish Society for the Study of Obesity (SEEDO)) [27] and periodontitis, interdental clinical attachment loss (CAL) detectable at ≥2 non-adjacent teeth, or buccal or oral CAL ≥3 mm with pocketing >3 mm detectable at ≥2 teeth, according to the definition of the 2017 World Workshop [3]. The following exclusion criteria were established: fewer than fourteen teeth, inflammatory oral diseases (infectious or otherwise); periodontal treatment in the previous six months or antibiotics in the previous three months; current systemic anti-inflammatory treatment; pregnancy or lactation; severe disease (including malignancies and alcohol/drug abuse); psychiatric disorders, any medical condition requiring antibiotic treatment before the intervention and a history of cardiovascular disease, chronic inflammatory disease or diabetes mellitus (diagnosed using the criteria of the American Diabetes Association criteria).

This pilot study carried out in humans was conducted in line with the ethical principles stated in the Declaration of Helsinki. All procedures carried out were approved by the Ethics Committee of our hospital (protocol number 36/14). All those participating, including the umbilical-cord donors, gave their written informed consent.

### 2.2. Interventions

The entire study population underwent non-surgical periodontal treatment. Obese patients were randomized via the minimization method using OxMaR (Oxford Minimization and Randomization) software. This method is a dynamic randomization technique that has been widely used in clinical trials for achieving a balance of prognostic factors across treatment groups. In our case, sex, age and BMI were considered prognostic factors and resulted in the formation of two groups: half of the participants (*n* = 26) underwent dietary therapy during the week that the periodontal study and non-surgical periodontal treatment were performed, while the other half (*n* = 26) did not undergo dietary intervention until the experimental period had ended, thus constituting a control group of obese subjects who did not lose weight.

The same clinician that performed the periodontal determinations (M-H, M.) was responsible for administering the non-surgical periodontal treatment; namely, instructions for oral hygiene and a full-mouth mechanical periodontal debridement performed with an ultrasonic device (Suprasson Newtron, Satelec, Acteon, Merignac, France) and gracey manual curettes (Hu-Friedy, Chicago, IL, USA). The full-mouth treatment (scaling and root planning) was performed in a single session, and the following adjunctive treatment was prescribed: use of a 0.12% chlorhexidine mouthwash (Bexident Encías^®^, ISDIN S.A, Barcelona, Spain) for 60 s, twice per day, for 14 days [28], in the absence of local and/or systemic antibiotics.

Dietary therapy consisted of a VLCD (very low-calorie diet) followed for six weeks. The VLCD consisted of a liquid formula (Optisource Plus^®^, Nestlé S.A., Vevey, Switzerland) with the following composition: 52.8 g protein, 75 g carbohydrates, 13.5 g fat and 11.4 g fibre, providing 654 kcal/day and all the essential vitamins, minerals and trace elements according to Recommended Dietary Allowances. After the six-week VLCD, subjects were assigned a low calorie diet for the following next six weeks: 50–55% carbohydrate, 28–33% fat and 15–20% protein, with an average daily energy intake of 1200–1800 kcal/day (recommended caloric requirements). Subjects were recommended to drink more than two liters of calorie-free liquids per day.

### 2.3. Determinations

All determinations were recorded at baseline and 12 weeks after intervention. In addition, we recorded data concerning medication patients were receiving at the time and smoking habit.

#### 2.3.1. Clinical Periodontal Determinations

A full-mouth periodontal examination was performed to measure clinical attachment loss (CAL), probing depth (PD) and gingival bleeding on probing (BOP) at six sites per tooth for all teeth, excluding third molars. PD was measured from the gingival margin to the base of the clinical pocket; CAL was recorded as the distance from the cement–enamel junction to the base of the clinical pocket, and percentage of BOP was calculated by dividing the number of sites with BOP by the number of sites explored and multiplying this value by 100. The average CAL and PD were calculated for each participant, as was the percentage of sites with BOP. In addition to these periodontal parameters, we recorded the number of teeth with PD ≥4 mm and the percentage of sites with PD 1–3 mm, PD 4–5 mm and PD ≥6 mm. Periodontal assessment was carried out using a conventional manual periodontal probe model PCP UNC-15 (Hu-Friedy, Chicago, IL, USA).

#### 2.3.2. Anthropometric and Biochemical Determinations

We calculated BMI by dividing the subject’s weight in kilograms by the square of his/her height in meters (kg/m^2^). With a metric tape, we measured waist circumference at the natural indentation between the 10th rib and the iliac crest. Blood pressure was recorded twice consecutively with an automatic sphygmomanometer (Omron M3, Kyoto, Japan). The formula (initial weight − final weight)/initial weight) × 100) was employed to calculate percentage of weight loss after the diet.

Following 12-h overnight fasting, venous blood samples were collected, and glucose, total cholesterol (TC) and triglycerides (TG) were evaluated in serum by an enzymatic method using specific kits (Ref. 3L82-40, Ref. 7D62 and Ref. 7D74, respectively) from Abbott Diagnostics (Abbot Park, IL, USA). HDL levels were obtained with a Beckman LX20 analyzer (Beckman Corp., Brea, CA, USA) using a direct method (HDL with Ultra HDL Reagent Ref. 3K33 from Abbott Diagnostics, Abbot Park, IL, USA). The Friedewald formula was used to calculate LDL cholesterol (LDLc) when TG were below 300 mg/dL. Insulin levels were measured by immunoassay with Architect Insulin Reagent Kit (Ref. 8K41) from Abbott Diagnostics (Abbot Park, IL, USA) and insulin resistance was calculated using the Homeostasis Model of Assessment (HOMA-IR = (fasting insulin (μU/mL) × fasting glucose (mg/dL)/405)). High-sensitive C-reactive protein (hsCRP) levels were quantified by an immunonephelometric assay with high sensitivity c-reactive protein reagent (Ref. 474630) from Beckman Coulter (Brea, CA, USA), RBP4 systemic levels were assessed by means of nephelometry assay with a RBP-800 kit Ref. TD-42661 from Trimero Diagnostics, (Barcelona, Spain), and serum levels of tumor necrosis factor alpha (TNFα) and P-selectin were determined with a Luminex^®^ 200 analyzer system (Austin, TX, USA) using MILLIPLEX^®^ MAP Human Cytokine/Chemokine and Cardiovascular Disease Magnetic Bead Panel kit, respectively (Millipore Corporation Billerica, MA, USA).

#### 2.3.3. Cell Isolation

Leukocyte PMNs were isolated by incubating citrated blood samples with dextran 3% for 45 min at room temperature. The resulting supernatant was then dropped over Ficoll-Paque Plus (GE Healthcare, Uppsala, Sweden) and centrifuged for 25 min at 650 g in order to isolate the PMNs and get rid of the remaining erythrocytes. After the pellet had been washed and suspended in Hank’s Balanced Salt Solution (HBSS) (Capricorn, Ebsdorfergrund, Germany), cells were counted with a Scepter 2.0 cell counter (Millipore Corporation, Billerica, MA, USA).

#### 2.3.4. Evaluation of Oxidative Stress Parameters

With the objective of evaluating oxidative stress parameters, we employed a life cell imaging method. For fluorescence determinations, PMNs were seeded in a plate with 48-wells (1.5 × 10^5^ cells/well) and incubated with different fluorescence probes diluted in HBSS for 30 min at 37 °C. Nuclei were visualized using the specific nuclear stain Hoechst 33,342. Fluorescence was visualized with a fluorescence microscope (IX81 Olympus). CellR software (Olympus, Shinjuku, Tokyo, Japan) was employed to capture individual images. The fluorescent signal was quantified individually (20 live cell images/well) by static cytometry software “ScanR” version 2.03.2 (Olympus, Shinjuku, Tokyo, Japan). Total ROS production was assessed with the 2′,7′-dichlorodihydrofluorescein diacetate (DCFH-DA) fluorochrome. Superoxide production by leukocytes was detected with dihydroethidium dye (DHE); levels of cytosolic Ca^2+^ were indicated by Fluo-4, and alterations in mitochondrial membrane potential were assessed using tetramethylrhodamine methyl ester (TMRM). All fluorescent probes were purchased from Life Technologies (Thermo Fisher Scientific, Waltham, MA, USA).

Antioxidant status was determined based on superoxide dismutase (SOD) activity in serum, as indicated by the manufacturer’s instructions (Superoxide Dismutase Assay Kit, Cayman Chemical, MI, USA).

#### 2.3.5. Adhesion Assay

A parallel plate flow chamber connected to an inverted microscope (Nikon Eclipse TE 2000-S) allowed us to measure leukocyte-endothelial cell interactions in vitro. Prior to this, we had collected human umbilical vein endothelial cells (HUVECs) from fresh umbilical cords donated by healthy donors. These HUVEC were seeded on coverslips at 1 × 10^3^ cells/mm^2^. Cells were cultivated to confluence in complete EMB-2 culture medium (Lonza, Basel, Switzerland), after which the coverslips were placed in the bottom plate of a flow chamber. One million leukocytes in 1 mL of RPMI medium (Gibco; Thermo Fisher Scientific, Waltham, MA, USA) were drawn across the HUVEC at a flow rate of 0.36 mL/min. A video camera (Sony Exware HAD; Koeln, Germany) connected to the microscope permitted a 5 × 25 mm view of the endothelial cells, and different leukocyte parameters—rolling velocity, rolling flux and adhesion—were evaluated over a period of 5 min, as we have described previously [23].

Platelet-activating factor (1 μM, 1 h) and TNFα (10 ng/mL, 4 h) (Sigma Aldrich, MO, USA) were employed as positive controls for the activation of PMNs and HUVECs, respectively.

### 2.4. Statistical Analysis

Based on preliminary data [23], this study was designed to have a power of 80% and to detect differences of ≥5 cells/mm^2^ between two paired means of the primary criterion of efficacy (minimum expected difference in leukocyte adhesion), assuming a common standard deviation of 7 units. Accordingly, 16 subjects were considered. Statistical analysis of the data was performed with SPSS 19.0 software (SPSS Statistics Inc., Chicago, IL, USA). Continuous variables in the tables were expressed as mean ± standard deviation (SD) for parametric data, and non-parametric data were expressed as the median and 25th and 75th percentiles. The figures show the mean + standard error (SE). To evaluate changes after the intervention, we used a paired Student’s *t*-test for parametric samples and a Wilcoxon test for non-parametric samples. Both groups of obese patients—one undergoing and the other not undergoing dietary therapy—were compared at baseline and at 12 weeks by means of an unpaired Student’s *t*-test for parametric samples or a Mann–Whitney U-test for non-parametric samples. Qualitative data were expressed in percentages and proportions were compared with a Chi-square test. A confidence interval of 95% was applied to all the tests, and differences were considered statistically significant when *p* < 0.05.

## 3. Results

A total of 52 patients with grade II-III obesity (BMI ≥ 35 kg/m^2^) and periodontitis were recruited for the study, three of whom eventually dropped out (they did not return at 12 weeks). All the remaining 49 subjects (32 women and 17 men; mean age: 43.7 ± 8.5 years) underwent non-surgical periodontal treatment; 23 of these did not undergo dietary therapy and 26 underwent dietary weight loss intervention.

### 3.1. Clinical and Biochemical Parameters

Table 1 presents anthropometric and biochemical parameters. At baseline, no significant differences in age, sex, or anthropometric or metabolic/inflammatory variables were detected between the groups. After intervention, anthropometric parameters in the dietary therapy group—body weight, BMI, and waist circumference—were found to have decreased significantly (*p* < 0.001), whereas HDLc increased (*p* = 0.025). Blood pressure (systolic and diastolic), total cholesterol, LDLc, tryglicerides and hydrocarbonated metabolism parameters (glucose, insulin and HOMA-IR) remained unchanged after weight loss. Of the total population, 30.6% of patients were taking antihypertensive drugs, and 22.4% were on antihyperlipidemic drugs, with no significant differences between groups (*p* > 0.05). A Chi-square test (*p* = 0.566) revealed no differences between the two groups of obese patients with regard to a smoking habit, with 77.6% of the participants proving to be non-smokers.

Systemic inflammatory parameters were altered by intervention. Specifically, TNFα levels decreased after dietary therapy (*p* = 0.029), while RBP4 levels were reduced in both groups after periodontal treatment (*p* = 0.043 in obese subjects undergoing dietary therapy and *p* = 0.014 in obese subjects not following the diet). However, hsCRP levels did not change after 12 weeks of treatment.

Regarding periodontal disease, in the dietary weight loss intervention (*n* = 26), 13 were in stage I (initial periodontitis), 11 were in stage II (moderate periodontitis) and 2 were in stage III (severe periodontitis). Among those that did not follow a diet (*n* = 23), 10 were in stage I, 8 in stage II and 5 in stage III. There were no differences in clinical periodontal parameters between the groups at baseline, as well as, no differences were observed in the periodontal parameters between the groups 12 weeks after the intervention (Table 2). As we had expected, a significant improvement in all clinical periodontal parameters was observed across the whole study population after non-surgical periodontal treatment. However, no differences were observed between the two groups in terms of the changes observed in periodontal parameters 12 weeks after the intervention.

### 3.2. Evaluation of Intracellular Leukocyte Parameters and Superoxide Dismutase Activity

ROS production, mitochondrial membrane potential and cytosolic Ca^2+^ in PMNs were determined by means of static cytometry. As a whole, after the intervention, we observed a significant decline in total superoxide production (*p* = 0.001) (Figure 1A) and lower levels of cytosolic Ca^2+^ (*p* = 0.006) (Figure 2A). In addition, these changes were observed independently of the diet as an adjuvant therapy (Figure 1B and Figure 2B and representative images in Figure 1C and Figure 2C, respectively).

Additionally, total ROS (Figure 1D) was significantly lower in PMNs from all subjects after intervention (*p* = 0.010). Although a downward trend was noted in both obese groups, the reduction did not prove to be significant when the two populations were clustered (Figure 1E). Furthermore, an increase in antioxidant defenses was manifested by significantly higher SOD activity in serum (*p* = 0.040), specifically after dietary weight loss intervention (*p* = 0.002) (Figure 1F,G). On the other hand, mitochondrial membrane potential was found to be unaltered twelve weeks after intervention, though there was a trend towards a decline in the patients who had undergone dietary therapy (Figure 2D,E).

### 3.3. Leukocyte-Endothelial Cell Interaction Assay

Finally, we investigated whether or not these changes—manifested mainly through leukocyte activation—were companied by an increase in adhesion under flow conditions, and to do this, we performed the assay under flow conditions. The results showed a statistically significant reduction in PMN adhesion in the whole population after intervention (*p* = 0.018) (Figure 3E). Nevertheless, when the groups were considered separately, this response was found to be specific to patients that had undergone dietary therapy (*p* = 0.019) (Figure 3F) and was associated with decreased levels of P-selectin in serum (126.1 ± 53.1 vs. 108.06 ± 43.27, *p* = 0.018). Neither PMN rolling velocity (Figure 3A,B) nor PMN rolling flux (Figure 3C,D) were altered twelve weeks after intervention.

## 4. Discussion

In this pilot study, an obese population of middle-aged subjects with periodontitis underwent non-surgical periodontal treatment in conjunction with, or in the absence of, dietary weight loss intervention. All clinical periodontal parameters improved subtly but in a statistically significant manner in both groups after periodontal treatment, whether or not the dietary program had been followed, without differences being observed between the groups. In addition, RBP4, total superoxide and intracellular Ca^2+^ were reduced in the whole population, independently of weight loss, suggesting that non-surgical periodontal treatment by itself improves leukocyte homeostasis. However, among the subjects that had followed the diet, a systemic reduction in TNFα was observed, together with an increase in HDL cholesterol and in enzymatic antioxidant systems such as SOD activity. These latter responses were associated with fewer adherences of leukocytes to the endothelium, which represented a reduced cardiovascular risk in these patients.

Under normal physiological conditions, a dynamic equilibrium is maintained between ROS and antioxidant defenses. Oxidative stress occurs when this equilibrium shifts in favor of ROS and is thought to play a causative role in the pathogenesis of many systemic diseases, including obesity and periodontitis. In periodontitis, oxidative stress has been described mainly in gingival tissues, saliva and gingival crevicular fluid [20,29]. Leukocytes are the main mediators of inflammatory response and are a key player in the onset of atherosclerosis.

Elevated ROS levels are a result of inflammation induced by neutrophils as they fight invading bacteria and play both direct and indirect roles in the destruction of periodontal tissue [4,5]. This situation is exacerbated in obesity, since circulating mononuclear cells are already in a proinflammatory state [30]. In this context, we have recently published data showing that PMNs from obese patients with periodontitis are hyperactive and increase cytoplasmic production of superoxide, and that this is associated with an alteration in leukocyte-endothelium cell interactions [8]. Our present findings take things a step further by illustrating the influence of non-surgical periodontal treatment on the same population; namely, a reduction of cytoplasmic levels of total ROS and total superoxide after periodontal treatment, in line with previous studies of the effect of non-surgical periodontal treatment on neutrophils of patients with periodontitis. In the study by Matthews et al., extracellular ROS production was reduced after therapy [31], while more recently Ling et al. reported that neutrophils of patients with periodontitis released less superoxide following two months of non-surgical periodontal treatment [7]. In addition, a local or systemic reduction of oxidative stress parameters such as 8-hydroxy-deoxyguanosine [26], reactive oxygen metabolites [32], total oxidative status, malondialdehyde and SOD [33] has been reported after non-surgical periodontal treatment, while a significant increase in SOD activity has been observed in platelets [34]. SOD is the major antioxidant enzyme responsible for superoxide removal, and it plays a critical role as a primary antioxidant defense in tissue. However, the findings of research about periodontal status and its relationship with systemic SOD activity are contradictory. While some authors have found SOD activity to be decreased after non-surgical periodontal treatment [33], others have observed a compensatory increase to levels similar to those of healthy subjects. Our data show that serum SOD activity was increased after periodontal treatment specifically in the subjects that underwent adjunctive dietary therapy, suggesting that the effect was owing to said diet.

The endoplasmic reticulum (ER) is an intracellular Ca^2+^ store of inflammatory cells and a key regulator of cell function [35]. Accumulating evidence associates a rise in intracellular free Ca^2+^ concentration in neutrophils with increases in chemotaxis, ROS release and apoptosis. In this regard, mediators of oxidative stress, apoptosis and Ca^2+^ entry are higher in the leukocytes of patients with pathologies such as multiple sclerosis, ankylosing spondylitis or obesity, and treatment with compounds with anti-inflammatory properties or weight loss have been shown to modulate these responses [36,37,38]. Our present data demonstrate that non-surgical periodontal treatment reduces abnormal Ca^2+^ distribution and ROS (mainly total superoxide), which would suggest a recovery of cell homeostasis. Further investigation is needed to evaluate to what extent these changes are implicated in the pathophysiology of periodontitis. By contrast, mitochondrial membrane potential does not seem to be involved in the underlying mechanisms of periodontitis, since it is unaltered by the presence/absence/grade of the disease, as we have previously published [8]. Moreover, as shown in the present study, mitochondrial membrane potential is not affected by non-surgical periodontal treatment. The fact that we have observed a downward trend after dietary weight loss intervention suggests that weight loss is capable of modulating mitochondrial function, in line with a recent report published by our group [36].

Past clinical studies have found a specific relation between elevated serum RBP4 levels and cardiovascular disease [15,39,40]. In endothelial cells, RBP4 treatment undermines mitochondrial function and increases vascular oxidative stress by inducing mitochondrial superoxide production [41], in accordance with the reduction in systemic RBP4 levels and the restoration of leukocyte homeostasis (mainly total superoxide production) we have observed in the present study. In addition, leukocyte-endothelial cell adhesion induced by TNFα leads to an increased production of ROS by said cells [42,43], which promotes leukocyte transmigration to the subendothelial space. Although there is increasing evidence that periodontal treatment can improve endothelial function [44,45,46,47], as far as we know, there are no reports to date about its effect in an obese population. Previous randomized clinical trials have evaluated the effect of periodontal treatment on vasodilation and subclinical arterial thickness in patients with periodontitis in the presence or absence of comorbidities. In fact, Tonetti et al. showed an improvement in braquial artery flow-mediated dilation 6 months after therapy [44], in line with the observations of D’Aiuto et al. in a type 2 diabetic population [45], while Saffi et al. did not find changes in patients with coronary artery disease 3 months after therapy [46]. In addition, periodontal therapy was found to have reduced carotid intima-media thickness in an aboriginal Australian population 3 months after treatment [47].

The present findings show less leukocyte adhesion to the endothelium in patients that followed the dietary weight loss program, in whom systemic TNFα and P-selectin levels were also reduced. hsCRP remained unchanged after the experimental period, which is compatible with the findings of previous studies [44,48]. In some studies, non-surgical periodontal treatment has been linked to a decrease in systemic TNFα levels in a non-obese population [49], but we observed a significant reduction of TNFα only when non-surgical periodontal treatment was combined with weight loss, possibly due to the continuously high levels of synthesis of TNFα by adipocytes and macrophages in obese individuals [50]. However, in the present study, we did not observe a greater periodontal response in the obese group undergoing dietary weight loss intervention, as we did in a previous work [24]. This was probably due to differences at baseline in BMI, mean PD and percentage of sites with PD 4-5 mm between the groups, as well as the reduced sample size. In addition, although the non-surgical periodontal treatment produced a statistically significant improvement in all clinical periodontal parameters in both obese groups, these changes were subtle, and the subjects still presented several sites with residual pockets (PD ≥ 4 mm) and bleeding on probing 12 weeks after therapy. This limited impact of periodontal therapy is in accordance with Lakkis et al.’s assessment of morbidly obese patients [51], suggesting that obesity is implicated in the impaired response to non-surgical periodontal treatment. In fact, we have previously reported [17] a smaller reduction in the number of teeth with PD ≥4 mm three months after non-surgical periodontal treatment in an obese vs. a non-obese population. Therefore, the persistence of residual periodontal inflammation could be due to the presence of a high level of chronic subclinical inflammation in the morbidly obese population. Most of the participants in the present study had mild-moderate periodontitis, in which case a mechanical scaling and polishing would probably have been sufficient to treat those with periodontal pockets of 4 mm. However, due to the already existing systemic inflammation in the obese population, we performed a full-mouth scaling and root planning in all our patients in order to achieve a greater reduction of periodontal inflammation. In line with this, a previous randomized, controlled trial in patients with periodontitis and hyperlipidemia showed that two and six months after treatment, serum proinflammatory cytokine levels and triglyceride levels were significantly lower in the group that had undergone a full-mouth intensive scaling and root planning than in the group that had undergone a standard cycle of supragingival mechanical scaling and polishing [52]. In another randomized controlled trial in which patients with type 2 diabetes and periodontitis were randomly assigned to a treatment group (scaling and root planning) or a control group (supragingival removal of plaque and calculus using ultrasound), it was observed that periodontal treatment significantly improved periodontal and metabolic parameters, whereas no improvement was observed in the control group [53]. In order to confirm the relationship between weight loss, systemic inflammation, oxidative stress, endothelial dysfunction and periodontal therapy outcomes, it is necessary to perform further longitudinal prospective studies and randomized controlled trials in larger patient samples with more severe periodontitis, with different degrees of BMI, and over longer periods of time. It is also important to assess different periodontal therapies (non-surgical periodontal treatment vs. standard buccal cleaning and polishing) and to incorporate a control group that does not undergo periodontal treatment and a group receiving only dietary therapy.

We should bear in mind certain limitations of the present study. The sample size was moderately small, though it was supported by sample size calculation, and the follow-up period was relatively short. Additionally, we did not include a control group without periodontal treatment, and the before-and-after experimental design is a type of non-experimental design that requires caution due to “threats to internal validity”. Regarding periodontal outcomes, we should mention that the non-surgical periodontal treatment we applied produced limited changes in periodontal parameters. We observed a persistence of sites with residual pockets and bleeding on probing, probably due to the mild periodontal status of our patients, which may have made the improvements produced by the therapy appear negligible. In terms of the methodology, the DHE fluorescence probe we employed to detect superoxide involves limitations that could have interfered with our findings; for example, nonspecific redox reactions may have confounded our DHE-superoxide determinations. Nevertheless, we should point out that our initial results were supported by the DCFH fluorescence probe we subsequently performed. Further research should explore a potential association between modifications in the intracellular signaling of leukocytes and their interaction with the endothelium. Moreover, the risk of subsequently developing atherosclerosis and cardiovascular disease requires further investigation.

## 5. Conclusions

To sum up, our results suggest that non-surgical periodontal treatment reduces total ROS (mainly superoxide) and promotes the recovery of cellular homeostasis in leukocytes of obese subjects, independently of whether there has been weight loss. When periodontal treatment is accompanied with dietary intervention, a reduction in systemic TNFα levels and an increase in SOD activity appear to mediate additional effects by which leukocyte-endothelium cell interactions and cardiovascular risk factors are reduced. Our data provide further knowledge about the pathogenesis of atherosclerosis and encourage the use of dietary therapy in conjunction with non-surgical periodontal treatment as a way of impeding the atherosclerotic process.

## Figures and Tables

**Figure 1 jcm-09-02117-f001:**
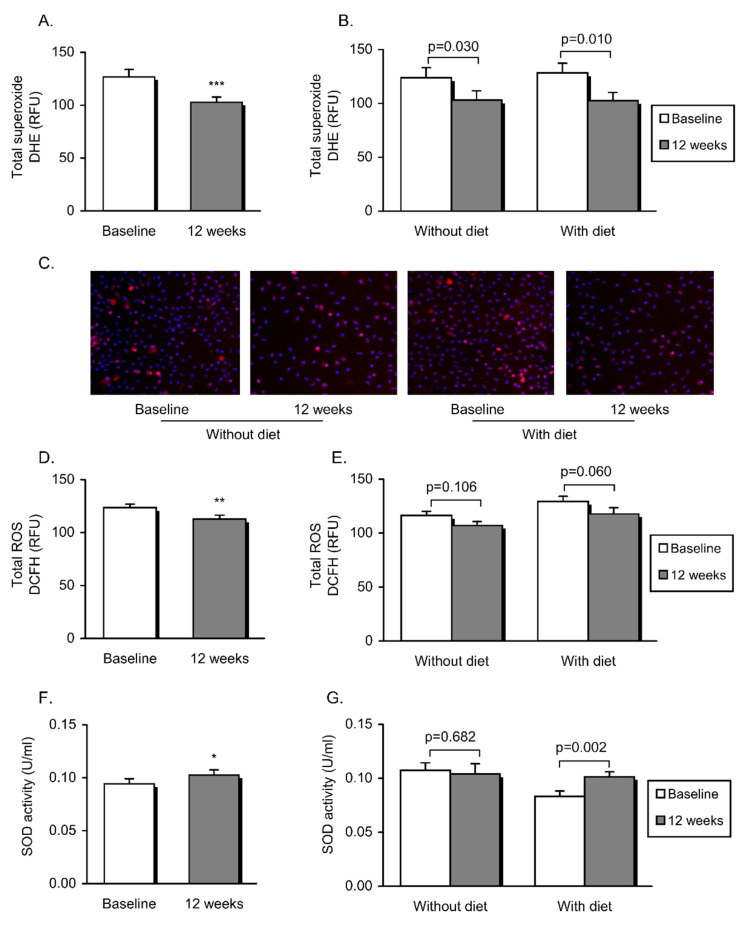
Parameters of oxidative stress in the study population before and after non-surgical periodontal treatment and dietary weight loss intervention. Total superoxide production was measured as arbitrary units of DHE fluorescence (**A**,**B**), and representative images depict DHE staining in red and Hoechst 33,342 nuclei fluorescence in blue (**C**). Total reactive oxygen species was calculated as the mean fluorescence intensity of DCFH (**D**,**E**). Superoxide dismutase activity in serum is also shown (**F**,**G**). Data are represented as mean + standard error. A paired Student’s *t*-test (* *p* < 0.05; ** *p* < 0.01; *** *p* < 0.001) was employed to compare the whole obese population before and after intervention. Obese patients undergoing vs. those not undergoing dietary therapy were compared before and after intervention using a paired Student’s *t*-test (differences were considered significant when *p* < 0.05).

**Figure 2 jcm-09-02117-f002:**
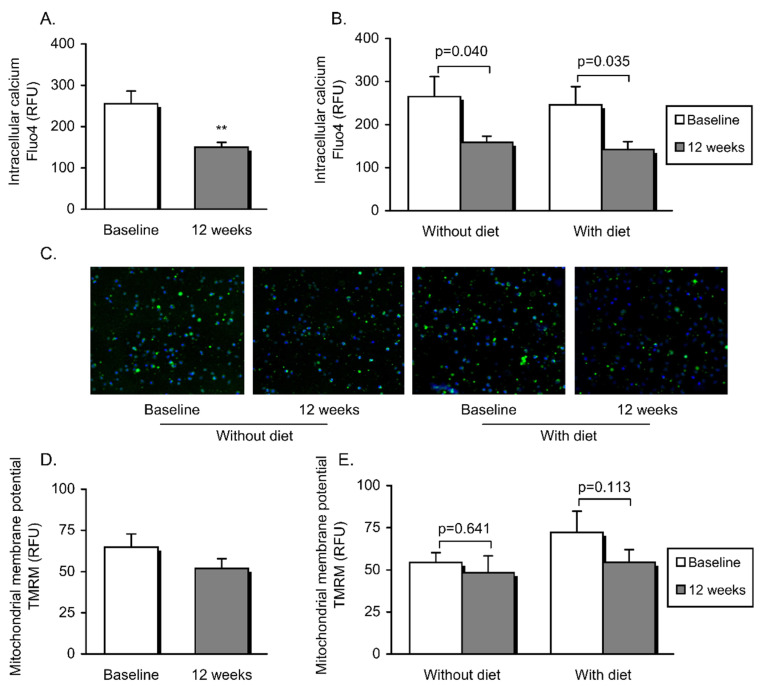
Mitochondrial function and cytosolic Ca^2+^ in patients’ leukocytes prior to and following non-surgical periodontal treatment and dietary weight loss intervention. Intracellular calcium was calculated as arbitrary units of Fluo4 fluorescence (**A**,**B**). Fluorescence microscopy images of leukocytes labelled with Fluo4 (green) and Hoechst 33,342 (blue) for nuclei staining (**C**). Mitochondrial membrane potential was measured as the mean of fluorescence intensity of TMRM (**D**,**E**). Data are shown as mean + standard error. A paired Student’s *t*-test (** *p* < 0.01) was used to compare the whole obese population before and after the intervention. Obese patients undergoing vs. those not undergoing dietary therapy were compared before and after intervention (also using a paired Student’s *t*-test; differences were considered significant when *p* < 0.05).

**Figure 3 jcm-09-02117-f003:**
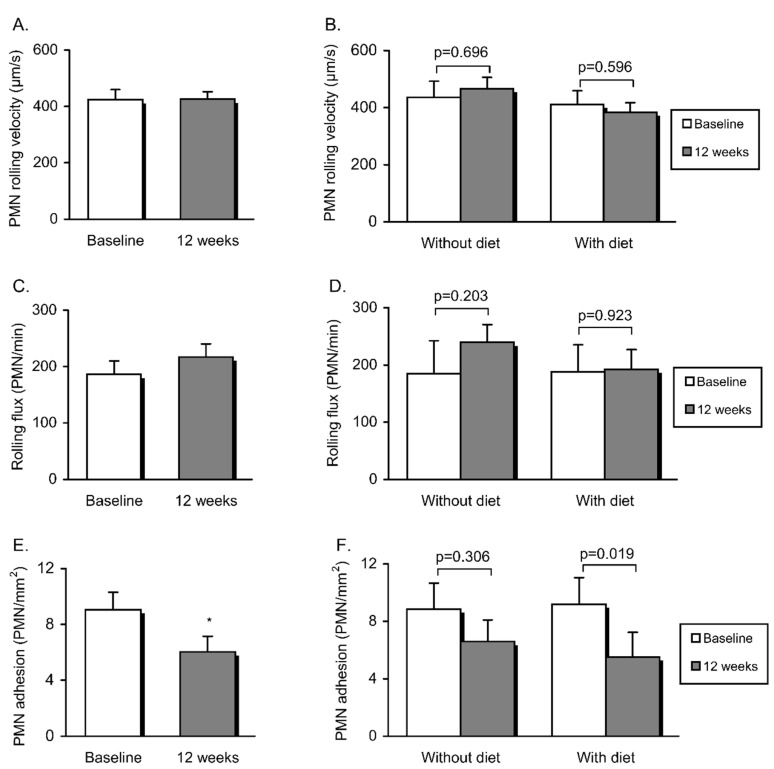
Endothelial function (determined by PMN-endothelial interactions) in the study population before and after non-surgical periodontal treatment and dietary weight loss intervention: PMN rolling velocity (**A**,**B**), PMN rolling flux (**C**,**D**) and PMN adhesion (**E**,**F**). Data are represented as mean + standard error. The whole obese population was compared before and after intervention using a paired Student’s *t*-test (* *p* < 0.05). Obese patients undergoing vs. those not undergoing dietary therapy were compared before and after intervention (also using a paired Student’s *t*-test; differences were considered significant when *p* < 0.05).

**Table 1 jcm-09-02117-t001:** Anthropometric parameters and biochemical variables in an obese population, some of which underwent adjunctive dietary therapy, at baseline and 12 weeks after non-surgical periodontal treatment.

	Obese without Diet		Obese with Diet		All Obese Subjects	
	Baseline	12 Weeks	Baseline	12 Weeks	Baseline	12 Weeks
*n* (% females)	23 (60.9)	------	26 (69.2)	------	49 (65.3)	------
Age	43.6 ± 7.8	------	43.8 ± 9.3	------	43.7 ± 8.5	------
Weight (kg)	118.6 ± 24.1	120.0 ± 24.9	125.5 ± 17.0	115.7 ± 16.3 ***	122.3 ± 20.7	117.7 ± 20.7 ***
Weight loss (%)	------	1.11 ± 3.10	------	−7.76 ± 4.66	------	−3.60 ± 5.98
BMI (kg/m^2^)	42.3 ± 6.8	42.7 ± 6.8	44.6 ± 4.8	41.1 ± 4.7 ***	43.5 ± 5.9	41.9 ± 5.8 ***
Waist (cm)	122 ± 17	120 ± 16	125 ± 13	117 ± 14 ***	124 ± 15	119 ± 15 ***
SBP (mmHg)	137 ± 16	135 ± 16	134 ± 18	133 ± 16	135 ± 17	134 ± 16
DBP (mmHg)	85 ± 13	85 ± 11	84 ± 9	81 ± 11	85 ± 11	83 ± 11
Glucose(mg/dL)	95 ± 10	95 ± 12	97 ± 12	95 ± 13	96 ± 11	95 ± 13
Insulin (μU/mL)	18.4 ± 9.4	18.3 ± 7.1	15.7 ± 7.6	16.3 ± 9.2	17.0 ± 8.5	17.2 ± 8.3
HOMA-IR	4.49 ± 2.41	4.39 ± 1.81	3.84 ± 2.01	3.95 ± 2.61	4.13 ± 2.23	4.15 ± 2.27
TC (mg/dL)	179 ± 37	184 ± 42	181 ± 38	184 ± 46	180 ± 37	184 ± 44
HDLc (mg/dL)	42.4 ± 10.5	44.4 ± 12.5	40.9 ± 10.9	43.9 ± 12.0 *	41.6 ± 10.6	44.1 ± 12.1 *
LDLc (mg/dL)	111 ± 30	113 ± 33	114 ± 32	117 ± 39	113 ± 31	115 ± 36
TG (mg/dL)	137 (78, 172)	113 (101, 150)	112 (88, 152)	110 (88, 152)	121 (86, 167)	111 (91, 151)
hsCRP (mg/L)	4.41 (1.84, 7.69)	4.09 (1.93, 11.15)	6.06 (3.43, 9.85)	6.03 (3.99, 9.84)	5.59 (2.63, 8.46)	5.92 (3.34, 10.3)
RBP4 (mg/dL)	4.38 ± 0.99	3.95 ± 1.06 *	3.76 ± 1.10	3.35 ± 0.98 *	4.05 ± 1.09	3.63 ± 1.05 **
TNFα (pg/mL)	17.8 ± 3.1	16.5 ± 4.01	17.1 ± 6.2	13.0 ± 1.8 *	17.4 ± 5.2	14.4 ± 3.3 **

Data represent mean ± standard deviation for parametric data and median (25th and 75th percentiles) for non-parametric data. * *p* < 0.05; ** *p* < 0.01; *** *p* < 0.001 when data of diet vs. no diet groups were compared at baseline and 12 weeks by a paired Student’s *t*-test for parametric data or by a Wilcoxon test for non-parametric data. Abbreviations: BMI: body mass index, SBP: systolic blood pressure, DBP: diastolic blood pressure, HOMA-IR: homeostasis model assessment of insulin resistance, TC: total cholesterol, HDLc: HDL cholesterol, LDLc: LDL cholesterol, TG: triglycerides, hsCRP: high sensitive C-reactive protein, RBP4: retinol-binding protein 4, TNFα: tumor necrosis factor alpha.

**Table 2 jcm-09-02117-t002:** Clinical periodontal parameters of obese subjects undergoing vs. not undergoing adjunctive dietary therapy at baseline and 12 weeks after non-surgical periodontal treatment.

	Obese without Diet (*n* = 23)			Obese with Diet (*n* = 26)				All Obese Subjects (*n* = 49)	
	Baseline	12 Weeks	Absolute Change	Baseline	12 Weeks	Absolute Change	*p*-Value Change	Baseline	12 Weeks
Mean PD (mm)	3.20 ± 0.52	3.02 ± 0.51 **	−0.18 ± 0.22	2.97 ± 0.43	2.79 ± 0.33 ***	−0.17 ± 0.16	0.884	3.07 ± 0.48	2.90 ± 0.44 ***
Mean CAL (mm)	3.26 ± 0.63	3.08 ± 0.61 **	−0.18 ± 0.20	2.97 ± 0.43	2.85 ± 0.36 **	−0.13 ± 0.22	0.443	3.10 ± 0.55	2.95 ± 0.50 ***
Teeth PD ≥4 mm (*n*)	19.6 ± 5.5	14.8 ± 5.9 **	−4.82 ± 6.01	18.2 ± 5.8	13.9 ± 6.16 ***	−4.38 ± 4.35	0.773	18.9 ± 5.6	14.3 ± 6.00 ***
Sites PD 1−3 mm (%)	63.5 ± 19.5	74.4 ± 21.5 **	10.8 ± 12.5	73.7 ± 15.9	82.5 ± 12.5 ***	8.76 ± 7.91	0.489	69.1 ± 18.2	78.8 ± 17.5 ***
Sites PD 4–5 mm (%)	32.3 ± 14.5	23.1 ± 17.7 **	−9.22 ± 12.53	24.6 ± 13.3	16.5 ± 10.3 ***	−8.11 ± 8.09	0.713	28.1 ± 14.3	19.5 ± 14.4 ***
Sites PD ≥6 mm (%)	4.13 ± 6.41	2.52 ± 4.41 *	−1.62 ± 2.92	1.68 ± 5.35	1.00 ± 3.07*	−0.68 ± 2.35	0.226	2.80 ± 5.92	1.69 ± 3.78 **
BOP (%)	29.1 ± 14.0	16.8 ± 10.6 ***	−12.3 ± 13.6	27.3 ± 14.6	17.0 ± 10.1 ***	−10.3 ± 10.9	0.573	28.1 ± 14.2	16.9 ± 10.2 ***

Data represented mean ± standard deviation * *p* < 0.05; ** *p* < 0.01; *** *p* < 0.001 when data of diet vs. no diet groups were compared at baseline and 12 weeks using a paired Student’s *t*-test. Obese without diet vs. obese with diet were compared at baseline and 12 weeks with an unpaired Student’s *t*-test. Absolute changes of periodontal variables in obese patients with or without dietary therapy were compared by an unpaired Student’s *t*-test.

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
