# Peer review of "Effect of Non-Surgical Periodontal Treatment on Oxidative Stress Markers in Leukocytes and Their Interaction with the Endothelium in Obese Subjects with Periodontitis: A Pilot Study"

_jcm, 2020, doi:10.3390/jcm9072117_

Round 1
Reviewer 1 Report
Comments to the Author:
This paper evaluates the effects of non-surgical periodontal therapy (NSPT) with or without dietary therapy on leukocyte parameters of oxidative stress and leukocyte-endothelial cell interactions in obese subjects. Whereas NSPT might induce an improvement in leucocyte homeostasis, dietary therapy reduces systemic inflammation and increases antioxidant status which, in turn, improve leukocyte function. The present study is based on several previous studies published by the authors. In 2017, the authors have already evaluated serum Retinol-Binding Protein 4 levels before and after periodontal therapy in lean and obese subjects with chronic periodontitis. In 2018, three papers have evaluated: 1) the relationship between oxidative stress parameters in leukocytes and leukocytes–endothelial cell interactions in obese patients with and without chronic periodontitis; 2) whether weight loss improves the response of obese subjects to non-surgical periodontal treatment; 3) whether dietary therapy intervention improves markers of oxidative stress in leukocytes and subclinical parameters of atherosclerosis.
Although the research hypothesis is interesting, the design and specific focus of the present study are rather confusing and need to be clarified. The uncontrolled before-And-after design here is not the most convincing and the results novelty in light of the previous work seems weak.
Specific Comments
- Title. The title should better reflect the primary objective, and include the study design of the study and only (Before-And-After).
- Abstract.
-The primary objective (effect of NSPT), and the secondary objectives (effect of dietary therapy), as well as the main study design should be clarified.
-The full name of RBP4 should be spelled out the first time.
- Introduction.
- The reference for the definition of periodontitis is not appropriate. We suggest to quote the definition from the last international Workshop consensus.
-The biological rationale for the biochemical measurements (especially RBP4), and the parameters of oxidative stress (especially mitochondrial membrane potential, Ca++) should be described in the context of periodontitis and cardiometabolic disorders.
-The research hypothesis, the primary objective (effect of NSPT), the secondary objectives (effect of dietary therapy), as well as the main study design should be clarified.
- Materials and Methods.
-Details on the patients’ recruitment are missing, and the inclusion criteria should be described, especially regarding the severity criteria of periodontitis.
-Details should be provided of randomization’s method.
- Results.
-No differences were observed in the periodontal parameters between the groups at 12 weeks post NSPT (Table 2). Absolute changes and p values should be presented in the table. These results seem contradictory with the authors’ previous work published in 2018 that concluded by suggesting a greater response to periodontal treatment in obese patients with dietary weight loss intervention.
Is there any explanation? These findings should be carefully interpreted in the discussion section.
-BoP are still above 10% after NSPT in both groups and a relatively high percentage of sites with PD > 4 mm in the perio without diet group meaning that periodontal inflammation was not resolved. This result should be discussed.
-Figure 2. In the Title, “cytolosic” should be edited.
- Discussion.
-Alternative study design such as RCT with a periodontal treatment group and a prophylaxis treatment group (hygiene and scaling only) should be discussed.
-Alternative methods for oxidative stress parameters and endothelial dysfunction measurements should be discussed.
-Limitations of the study should be added, especially regarding the design (absence of a control group), the quality of the periodontal treatment clinical results (BoP), the short-time follow up.
Reviewer 2 Report
- A title change should be considered. The number of patients is very low. It would be advisable to assume it as a pilot study.
- The authors should clarify the following methodological aspects:
- Type of tubes used for blood collection.
- Amount of blood drawn.
- Specify the commercial kits used for the determinations, For example, which kit was used for the Luminex analyzes? for insulin immunoassay? etc.
- Line 152. Temperature of centrifugation
3. The authors should clarify why the two groups were compared together, at baseline and after 12 weeks. Is diet considered to have no effect?
Round 2
Reviewer 1 Report
The authors have perfectly considered all my comments.
Here, are some minor edits:
-line 96: the “(“ before “interdental” should be deleted.
-line 97: the definition of a periodontitis case must be modified as follows:”…by buccal or oral CAL ≥ 3 mm with pocketing > 3 mm…” (NOT ≥ 3 mm) and the reference n°28 changed for the ref n°3 (Tonetti et al J Clin Perio 2018)
